# Analyzing the Energy and Damage Constitutive of Cemented Backfill with Different Water Content under Dynamic Load

**DOI:** 10.3390/ma16165677

**Published:** 2023-08-18

**Authors:** Yu Hu, Zhuo Li, Yawen Su, Yongbo Wu, Xiaoshuai Li, Wenxue Gao, Xiaojun Zhang

**Affiliations:** 1Faculty of Architecture, Civil and Transportation Engineering, Beijing University of Technology, Beijing 100124, China; HUyu0420@emails.bjut.edu.cn (Y.H.); zhangxj@bjut.edu.cn (X.Z.); 2The Seventh Metallurgical Construction Co., Ltd., Guiyang 550014, China; m15585288600@163.com

**Keywords:** cemented filling body, moisture content, energy evolution, damage characteristics, constitutive model

## Abstract

The dynamic characteristics of the filling body are the key parameters for designing the filling ratio and evaluating the stability of an underground stope. The different environment (water-bearing state) of the filling body in the underground stope exerts a complex impact on the mechanical behavior of the filling body. Therefore, six groups of cemented filling body specimens with different states were formed and subjected to dynamic uniaxial impact tests. The effects of water content on the mechanical properties, fractal dimension, and deformation damage characteristics of the cemented backfill under dynamic load were analyzed in depth, and a dynamic damage constitutive model that considers water damage and the compaction stage was established. The results indicate the following: (1) Due to the change of the specimen from the dry state to the water saturation state, the dynamic compressive strength of the cemented filling body decreases from 5.03 Mpa to 1.79 Mpa; however, the ductility of the specimen generally increases, and the filling body specimens with different water contents mainly exhibit tensile failure. (2) There is a significant nonlinear relationship between the water content and the fractal dimension *D_b_* of the cemented backfill specimen, and the growth rate of the fractal dimension *D_b_* tends to slow down with the increase in the water content. (3) From the energy evolution perspective, the water content of the specimen exerts a significant effect on the elastic deformation and failure stage of the stress–strain curve, and the slope of the dissipated energy-strain curve decreases with the increase in water content. (4) Based on the Weibull distribution and damage theory, a statistical damage constitutive model of cemented backfill was established, and it was compared with the experimental curve to verify the rationality of the model. Therefore, the relationship between stress and damage and the strain curves is discussed, and it is inferred that the damage evolution curve of cemented backfill is a typical *S*-shaped curve that exhibits a stable development-rapid increase-tending to be gentle. This study can provide a theoretical reference for further understanding the dynamic behavior and stability of backfill under different water conditions.

## 1. Introduction

Tailings are a typical mineral processing waste. Over the years, a large amount of tailings have been stacked in tailings ponds, posing a critical threat to the ecological environment and the safety of residents [1,2]. Due to the rapid increase in mining volume worldwide, how to dispose tailings safely and economically has become a crucial problem. Especially in the context of the current depletion of shallow resources, many local and foreign mines have gradually shifted to deep mining [3]. Deep mining is subjected to problems such as high ground stress, high rockburst, and high karst water pressure, and the large-scale application of tailings cemented filling is a potentially effective solution to these problems. Usually, the cemented tailings filling material mainly comprises aggregates (e.g., tailings and waste rock), a gel material, and water or an admixture. To minimize the impact of tailings, scientists worldwide have conducted relevant research on cement materials based on industrial waste and applied them to the backfilling of underground mines [4]. Because the filling mining method can provide a relatively safe working environment for staff, it has gradually developed into a mainstream mining method for the recovery of mineral resources in deep mines [5,6]. Meanwhile, in the underground mining process, whether the stope can be safely and stably mined is immensely affected by the stability state of the cemented filling pillar. The cemented filling body will not only be affected by the dynamic disturbance occasioned by the mining process, but also by the long-term seepage of groundwater [7]. Therefore, the systematic analysis of the dynamic characteristics and deformation characteristics, and the constitutive relationship between the filling body and different water contents can provide some theoretical support for the subsequent deep mining.

With regard to the research on the mechanical properties of cemented backfill, researchers have performed a considerable amount of studies. For example, relevant scholars [8,9] indicate that the activation treatment of tailings before the mixing of tailings and cemented dry materials can effectively enhance the strength and rheological properties of tailings. Yang et al. [10] conducted a uniaxial impact test on the high-concentration cemented filling body using the split Hopkinson bar system, and analyzed the relationship between the dynamic compressive strength, dynamic strain, strength enhancement factor, specific energy absorption, and average strain rate of the specimen. Zhang et al. [11] utilized the split Hopkinson bar system to analyze the dynamic mechanical properties and deformation, and the failure laws of layered backfill under dynamic loading conditions with 10–80 s^−1^ strain rates. It was observed that the dynamic compressive strength and dynamic strength growth factor of layered backfill were positively correlated with the strain rate. Meanwhile, the instability conditions of layered backfill were derived based on the Stenerding–Lehnigk criterion, which exhibited satisfactory agreement with the experimental results. Xue et al. [12] analyzed the dynamic mechanical properties of cemented tailings backfill (CTB) under different strain rates and spectral amplitudes. Ouattara et al. [13] evaluated the effects of different types of water reducers on the consistency and unconfined compressive strength of cemented paste backfill. Song et al. [14] analyzed the influence pertaining to the modification of the loading rate and the environment (i.e., humidity and temperature) of the filling body on the mechanical behavior (i.e., failure simulation, energy evolution, and damage characteristic curve) of the filling body. Unfortunately, there is no further quantitative analysis of the relationship between water content and mechanical parameters, such as dynamic compressive strength.

Currently, research on the effect of water content on the dynamic characteristics of materials mostly focuses on rock materials (Jin) [15]. Using indoor SHPB equipment, the different water content states of red sandstone are utilized as the research object, and the energy dissipation in the impact process is discussed in depth. Kim et al. [16] conducted static, medium strain rate, and high strain rate loading on saturated dry red sandstone and light yellow sandstone, respectively. It was observed that the static and dynamic compressive strength of the two sandstones in the saturated state decreased by approximately 20% compared with the corresponding strength in the dry state. The Young’s modulus of light yellow sandstone in the saturated state is lower than that in the dry state, whereas the Young’s modulus of red sandstone in the saturated state and dry state does not vary significantly. Lu et al. [17] conducted impact tests on six groups of sandstones with different water contents. It was observed that as the water content increased from 0% to 2.58%, the strength, dynamic elastic modulus, and unloading modulus of sandstone decreased. By analyzing the energy transmitted by the stress wave, it was noted that as the water content increased, the reflected wave energy increased, whereas the transmitted wave energy and the total energy dissipated by rock damage decreased. Daraei et al. [18] discussed the failure mechanism and dynamic characteristics of different rock specimens corresponding to different water content states. The results indicated that the static and dynamic compressive strength of sandstone decreased with the increase in water content. In addition, some scholars, who utilized the energy and load cycles perspectives, have proposed relevant modified constitutive models for rocks, and they obtained some useful conclusions [19]. The aforementioned research conclusions indicate that research on the dynamic mechanical properties of cemented backfill under different water contents should be further analyzed, and the damage constitutive model of cemented backfill under different water content conditions has become an urgent task in rock mass engineering.

Based on the aforementioned research, it can be noted that the analysis of the dynamic behavior and damage constitutive model of cemented backfill under different water contents is a prominent research topic. Therefore, this study aims to reveal the mechanical properties of cemented backfill materials with different water contents under dynamic disturbance scenarios, and to provide a theoretical basis for further understanding the mechanical behavior and stability of backfill corresponding to different water content conditions under dynamic load occasioned by mining disturbance. Thus, this study analyzes the mechanical properties (i.e., stress–strain behavior, failure mode, fractal dimension, energy dissipation, and damage evolution characteristics) of cemented backfill under different water-bearing states subjected to dynamic load and establishes a dynamic constitutive model that considers the influence of water damage.

## 2. Test Method

### 2.1. Test Materials

Ordinary Portland cement (P.032.5) was utilized as the gel material for the test specimen; laboratory tap water was utilized as the mixing water; and all tailings obtained from a mine in Yunnan Province, China were utilized as the filling aggregate. The density of the test tailings was 3.14 g/cm^3^. After drying, the particle size distribution range of the tailings was tested using a laser particle size analyzer (Figure 1). The *d*_10_ of the tailings was 34.47 μm, and the *d*_50_ was 65.26 μm. According to the tailings grading standard, it can be noted that the tailings utilized in the test are fine-grained tailings, which exhibit a large specific surface area and a relatively viscous slurry. The chemical composition of the tailings samples was analyzed, and Table 1 shows the analysis results. From the table, the main component of the tailings is SiO_2_, accounting for 37.15%, and CaO is the secondary component, accounting for 15.06%. Figure 2 depicts the XRD diffraction pattern test results of the tailings utilized in the test; it can be observed that the main crystalline peaks of the graded tailings obtained from a mine in Yunnan are quartz, lime, MgO, and bauxite.

### 2.2. Specimen Preparation

According to the commonly utilized filling ratio parameters of the mine, the lime–sand ratio of the filling body sample is set to 1:4, and its mass concentration is approximately 66%. The preparation process of the specimen is as follows: (1) Weigh the quantitative tailings and cement materials, and mix them evenly. (2) Add quantitative water, stir the mixture in the JJ-5 planetary cement mortar mixer for 5 min until the slurry is evenly mixed and the corresponding speed is 60 r/min; pour the uniformly mixed filling slurry into a cylindrical abrasive tool (the size of the abrasive tool is 50 mm × 25 mm according to the sample size selection method recommended by the International Society of Rock Mechanics ISRM [20]). (3) After 24 h of initial setting, the slurry was placed in a constant temperature and humidity curing box (temperature of (20 ± 1) °C, humidity of 95%) for 28 days [21,22]. The sample preparation results are shown in Figure 3. To form cemented backfill specimens with different water contents, the specimens were first completely dried; subsequently, the dried sample was subjected to water saturation treatment, and, finally, the saturated samples were baked to vary their moisture contents. The process of determining the water content of cemented backfill is as follows: First, the average water content of the sample in the saturated state was calculated using Formula (1). The average mass of the samples at the 20%, 40%, 60%, 80%, and 100% water contents was calculated using Formula (2). Finally, the saturated filling body specimens were placed in a matching oven for baking, and the heating temperature was set to 50 °C. The samples were weighed every 10 min, and their average mass *m* was recorded. The samples pertaining to different water content grades were wrapped with a plastic wrap, sealed with tape, and placed in a fixed curing box.
(1)$$\omega =\frac{m_{f}-m_{d}}{m_{d}}\times 100\%$$
(2)$$m_{c}=m_{d}+s*\omega$$

In the preceding formula, *m_c_* denotes the filling body quality under different water contents, *m_f_* denotes the saturated quality of the specimen, *m_d_* denotes the drying quality of the specimen, *s* denotes the moisture content grade, and *ω* denotes the water content of the saturated sample.

### 2.3. Experimental Devices

A separate SHPB device loading test system with a 50 mm diameter was utilized. Figure 4 depicts the equipment utilized in this test. The bar size is marked in the diagram. Notably, to maintain the end face of the specimen flush with all the bars in the test, lubricant was smeared on both sides of the specimen in advance; thus, the experimental error was reduced [23].

According to the basic stress wave theory, the voltage signal obtained using the test monitoring can be processed into a strain value, and the transformation relationship of the stress, strain, and strain rates with time in the specimen satisfies Formulas (3)–(5) [10].

In the formulas, *A_e_* and *A_S_* denote the end face area of the bar and specimen, respectively; *L_s_* denotes the specimen thickness; *C_e_* denotes the wave velocity; *E_e_* denotes the rod elastic modulus; and $\varepsilon_{I}(t)$, $\varepsilon_{R}(t)$, and $\varepsilon_{T}(t)$ denote strain-related signals.
(3)$$\sigma (t)=\frac{A_{e}E_{e}}{2A_{s}}\left[\varepsilon_{I}(t)+\varepsilon_{R}(t)+\varepsilon_{T}(t)\right]$$
(4)$$\varepsilon (t)=\frac{C_{e}}{L_{s}}\int_{0}^{t}\left[\varepsilon_{I}(t)-\varepsilon_{R}(t)-\varepsilon_{T}(t)\right]dt$$
(5)$$\dot{\varepsilon }(t)=\frac{C_{e}}{L_{s}}\left[\varepsilon_{I}(t)-\varepsilon_{R}(t)-\varepsilon_{T}(t)\right]$$

### 2.4. Test Scheme

Before the start of the SHPB test, it is necessary to conduct a pre-test under a certain loading pressure. Therefore, in the preparation stage of each impact test, the position of the bullet punch is pre-fixed, and the impact pressure is maintained at 0.25 MPa. According to the GB/T50123-2019 standard [24], the dynamic uniaxial compression test of cemented tailings backfill samples with different states at a 28 d curing age was performed using SHPB equipment. Before the test, the two ends of the sample should be ground and smoothed, and, subsequently, placed between the incident bar and the transmission bar. Herein, the cemented filling body specimens were set up in six different water-bearing states (approximately 0%, 20%, 40%, 60%, 80%, and 100%). The conditions for obtaining high reliability test results are a large number of experiments. To obtain the exact value closest to the actual value, seven samples of different water-bearing grades were formed, and the average value was utilized in the subsequent analysis.

## 3. Analysis of SHPB Test Results

### 3.1. Stress–Strain Behavior of Cemented Backfill under Different Water Contents

From Figure 5, it can be inferred that the stress–strain curves of the six water-bearing cemented backfills set in this experiment basically include four stages: compaction, linear elasticity, plastic deformation, and post-peak failure [25]. It should be noted that the filling body specimen did not fail immediately after attaining the peak stress. After the micro-crack gap of the undamaged part was squeezed and reduced, it was completely destroyed after the low stress peak appeared, which was consistent with the test results of related studies [26]. 

We utilize the stress−strain curve of the cemented backfill specimen with an 80% water content (Figure 6). In the fracture compaction stage (OA), the specimen was compacted due to its initial defects; the curve, herein, indicates a slightly concave shape, and the state of the specimen exerts a minimal effect on this stage. In the linear elastic stage (AB), the slope of the curve remains constant. By comparing the curves of the specimens in different states, we observe that when the state of the specimen changes from dry to saturated, the elastic modulus gradually decreases, which indicates that water will weaken the anti-deformation ability of the cemented backfill [27,28]. In the plastic deformation stage (BC), irreversible plastic failure occurred inside the filling body, and microcracks were continuously generated, expanding in the filling body and resulting in continuously accumulating damage. During this stage, as the strain increased, the slope of the curve began to decrease until the peak was zero. The peak stress was lowest in the saturated filling body specimen and highest in the dry-state specimen. Although the strain continued to increase in the post-peak failure stage (CD), the stress value began to decrease until a certain value was attained. During this stage, the cracks inside the sample converged, nucleated, and formed the dominant crack, which led to the failure of the bearing structure of the sample and the loss of bearing capacity. During this stage, the stress of the filling body in different water-bearing states decreased slowly, and the stress of the dry filling body rapidly dropped, which corresponds to a marked brittleness. Generally, with the increase in water content, the ductility of the specimen also increases gradually, and the aforementioned observation can be rationalized as follows: the existence of water can simultaneously weaken and lubricate the filling body specimen. Thus, the internal micro-cracks of the specimen are uniform and rapid, which can effectively disperse the stress, and subsequently avoid the sudden outbreak of energy accumulation and direct failure. Therefore, for the stress–strain curve (i.e., after the end of the elastic stage of the filling body specimen), the specimen with the higher water content will exhibit a longer plastic deformation stage.

### 3.2. Variation Characteristics of Dynamic Compressive Strength and Failure Mode with Water Content

The dynamic compressive strength (*σ_d_*) is generally utilized to characterize the ultimate bearing capacity of cemented backfill. Figure 7 indicates that the *σ_d_* value of the backfill specimen is exceedingly affected by the water content. Specifically, when the water content ω increased from 0% to 100%, the specimen value decreased from 5.03 MPa to 3.38 MPa, 2.72 MPa, 2.53 MPa, 2.05 MPa, and 1.79 MPa, and the corresponding reductions were 32.8%, 19.5%, 7.0%, 19.0%, and 12.7%, respectively. The quadratic function was utilized to fit the scatter plot according to the change rule illustrated in the figure, and a mathematical expression for the relationship between the value and water content ω was subsequently obtained. Compared with the specimen that exhibits a water content of 0, the value of the specimen with a 20% water content exhibited the most apparent decrease, attaining 32.8%. When the water content increased from 20% to 100%, the overall decrease in the value did not exceed 20%, and the fitting curve revealed that when the water content (ω) of the specimen increased, the slope of the fitting curve gradually decreased and tended to be gentle. 

Figure 7 also illustrates the failure mode and the filling body specimens with different water contents under impact dynamic loads from a macroscopic perspective. When the water content ranged from 0% to 20%, although the cemented backfill sample was destroyed, the main bodies were still retained, which indicates that it exhibited some bearing capacity, and no instability failure occurred. When the water content was in the 40–100% range, the cemented filling body was completely destroyed, no main body retention occurred, and the cemented filling body was destabilized. From the final fracture state of the specimen, the integrity of the cemented backfill structure after dynamic compression gradually decreased when the state of the specimen changed from dry to water-saturated. Moreover, the number and size of fragments also decreased, and the proportion of the pulverized powder particles gradually increased, which indicates that water will deteriorate the overall strength of the cemented backfill. The fragments after the failure of the cemented filling body were mainly columnar. The macroscopic failure surface formed after the failure of the specimen was consistent with the dynamic loading direction, which indicates that when the cemented filling body is under different water contents, it mainly exhibits the axial tensile failure mode. This observation is attributable to the following phenomenon: due to the Poisson effect, the cemented filling body produces large tensile stress under impact load, which results in tensile failure.

### 3.3. Fractal Analysis on a Cemented Filling Body with Different Water Contents

The fractal dimension is a crucial index for characterizing the fractal characteristics, and the index increases with the increase of the degree of fragmentation exhibited by the specimen. Many scholars have noted that the evolution and distribution of internal cracks in materials under dynamic loading satisfy the statistical self-similarity, and the number of fragment–particle sizes satisfy the following basic assumptions [29]:(6)$$\{\begin{array}{l} D=\frac{\ln N(r)}{\ln1/r}\\ N(r)\propto r^{-D} \end{array}$$

In the preceding formula: *D* denotes the fractal dimension, and *N*(*r*) denotes the number of fragments with a particle size greater than *r*.

According to the mass-equivalent size relationship that characterizes fractal geometry, the distribution equation that expresses the fragmentation of the specimen after impact crushing is as follows:(7)$$y=\frac{M_{r}}{M_{T}}={\left(\frac{r}{r_{m}}\right)}^{3-D_{b}}$$

In the preceding formula; r denotes the size of the fragment; *r_m_* denotes the maximum size of the fragment (mm); *M_r_* denotes the cumulative mass of the fragment size less than (g); *M*_T_ denotes the total weight of fragments (g); and *D_b_* denotes the fractal dimension. By substituting logarithms into both sides of the formula, we can obtain the following equation:(8)$$\mathrm{lg}(y)=\mathrm{lg}\left(\frac{M_{r}}{M_{T}}\right)=\left(3-D_{b}\right)\mathrm{lg}\left(\frac{r}{r_{m}}\right)$$

According to the preceding formula, the slope of the fitting line is (3 − *D_b_*).

According to the preceding formula, the relationship between the fractal dimension pertaining to the mass of the test pieces and the fractal dimension of the equivalent particle size can be calculated. As depicted in Figure 8, the slope distribution that indicates the fractal dimension pertaining to the mass of the test pieces and the fractal dimension of the equivalent particle size under different water contents is different. The slope decreases with the increase in water content, and when the water content is large, the slope becomes small. Table 2 and Figure 9 indicate that with the increase in water content, the increase rate of the fractal dimension *D_b_* value slows down, and there is a significant nonlinear relationship between the two factors. Specifically, when the water content rises from 0% to 60%, the fractal dimension *D_b_* increases from 2.39 to 2.55; subsequently, as the 60% water content approaches 100%, the corresponding *D_b_* increases from 2.55 to 2.61, and the increase rate decreases.

### 3.4. Energy Consumption Characteristics and Damage Evolution Analysis

The compression deformation of the filling body specimen occasioned by the dynamic load primarily entails the absorption and conversion of energy. Some of the energy will be stored in the specimen until the stress peaks, after which it is released; the remainder is consumed in the compaction of the initial specimen defects and in the development and expansion of new cracks. It is assumed that the effect of external factors during the test can be ignored. In addition, only the axial load acts under uniaxial compression. The effect of external factors can presumably be ignored during the test. The formulas pertaining to the strain energy (*U*), releasable elastic strain energy (*U_e_*), and dissipated energy (*U_d_*) of the cemented backfill are expressed as follows [30,31].
(9)$$U=\int_{0}^{\varepsilon }\sigma d\varepsilon$$
(10)$$U=U_{e}+U_{d}$$
(11)$$U_{e}=\frac{1}{2E_{0}}\sigma^{2}$$

In the formula, *E*_0_ denotes the initial elastic modulus of the specimen. 

Figure 10 plots the strain–strain energy curves of six groups of cemented backfill under different water contents, and similar to Figure 6, it divides the curve into the following four change processes: 

(1) In the first stage, the specimen was subjected to crack compaction, and the total strain energy curve coincided with the elastic strain energy curve. (2) When the specimen began to undergo elastic deformation in the second stage, the strain continued to increase, and the growth rate pertaining the elastic strain energy curve of the specimen increased correspondingly. The internal pores of the specimen were compacted to reveal the linear elastic properties of the solid material, and the strain energy was stored per unit strain increase. The influence of the specimen state in this stage was relatively small. (3) In the third stage, the specimen began to transit into the plastic deformation stage, during which new cracks appeared and expanded. The growth rate of elastic strain energy tended to slow down, whereas the dissipation energy maintained an approximately steady increase. The cemented filling body transited into the unstable crack expansion stage. The growth trend of elastic strain energy gradually decreased, and the elastic strain energy attained the maximum value during this stage. The dissipation energy steadily increased and the growth rate increased, which indicates that plastic deformation and damage begin at this stage and accumulate. In addition, the growth rate of the low-drying specimens was fastest in this stage, which indicates that the specimen exhibited high brittleness: as the ω value of the specimen increased, the rate at which the specimen transited into this stage gradually decreased. (4) In the fourth stage, the filling body sample transited into the failure stage. In this stage, the elastic strain energy decreased rapidly, whereas the dissipation energy curve increased sharply, which can effectively explain the marked internal damage of the cemented filling body. If the analysis in Figure 8 is considered, although the elastic strain energy of the specimen will be rapidly released in this stage, it will not always decrease to zero. From a macroscopic perspective, the bearing capacity of the specimen decreases and the residual strength of the material is low: the dissipation energy rapidly increases until the specimen is destroyed. 

Figure 10 confirms that at this stage, the dissipation energy curve of the dry and low-water-content filling body specimens increased almost linearly and the specimen was rapidly destroyed after peaking, which is indicative of strong brittleness. However, the slope pertaining to the dissipation energy curve of the filling body with a higher water content was relatively low. The aforementioned analysis indicates that although water weakens the filling body, the increase in ω enhances the ductility of the filling body: the cemented filling body with an appropriate water content will not be quickly destroyed under a certain strength dynamic load. Based on the preceding analysis, although water weakens the strength of the filling body, the increase of the ω value of the specimen also enhances the ductility of the filling body: the cemented filling body with the appropriate water content will not be rapidly destroyed under the impact dynamic load of a certain strength. After the failure, the specimen still retains a certain bearing capacity, which provides a valuable response time for reinforcement measures or personnel evacuation, and enhances the safety of the filling area.

## 4. Constitutive Model of Cemented Filling Body

### 4.1. Fundamental Assumption

It is generally believed that rock materials contain some initial defects with a disordered distribution and are composed of microelements, which can be regarded as particles. Meanwhile, according to Shan et al., the rock specimen can be regarded as the parallel connection of the damage body *D* and the viscous body *η* [32].
(12)$$\varphi (F)=\frac{m}{a}{\left(\frac{F}{a}\right)}^{m-1}e^{-{\left(\frac{F}{a}\right)}^{m}}$$

The strength of the microelement inside the rock material satisfies the Weibull distribution, and its density function is expressed as follows:(13)$$\sigma =\eta \frac{d\varepsilon }{dt}$$
where *F* denotes the distribution variable of the microunit strength.

The constitutive relation of viscous body *η* is formulated as follows.

The ratio of the number of failed elements to the total number of elements in the material *D_M_* is expressed as follows:(14)$$D_{M}=\frac{N_{\varepsilon }}{N}=\frac{\int_{0}^{{}_{\varepsilon }}NP(x)}{N}=1-e^{\left[-{\left(\frac{F}{m}\right)}^{m}\right]}$$

Macroscopic phenomenological damage mechanics proposes that the degree of material degradation under an external load can be reflected by the change in its basic physical properties. Accordingly, the elastic modulus *E_w_*, which can fully reflect different water contents, is selected; thus, the damage variable *D_w_* is defined as follows:(15)$$D_{w}=1-\frac{E_{w}}{E_{0}}$$

### 4.2. Coupling Damage under Hydration–Impact Load

The material damage variable considering the coupling of hydration and impact load can be obtained as follows based on the generalized strain equivalence principle [33]:(16)$$D=D_{M}+D_{w}-D_{M}D_{w}$$

From Formulas (14)–(16), the damage variable under hydration impact can be defined as follows:(17)$$D=1-\frac{E_{w}}{E_{0}}e^{\left[-{\left(\frac{F}{m}\right)}^{m}\right]}$$

According to the Lemaitre equivalent stress theory [34], the mathematical equation of the material constitutive can be expressed as follows:(18)σ=Eε(1−D)

Due to the parallel connection between the damaged body and the viscous body, the stress–strain relationship satisfies the following formula:(19)$$\{\begin{array}{c} \sigma =\sigma_{a}+\sigma_{b}\\ \varepsilon =\varepsilon_{a}=\varepsilon_{b} \end{array}$$

According to the existing literature [35], the quantitative relationship between the elastic modulus and water content can be expressed as follows:(20)$$E_{w}=E_{0}e^{aw}$$

In the formula, *E*_0_ denotes the elastic modulus of the dry specimen, *E_w_* denotes the elastic modulus of a specimen with a certain moisture content, ω denotes the water content, and *a* denotes the fitting coefficient.

Combining (15)–(17) into (14), the constitutive relationship of cemented backfill under different water contents is obtained as follows:(21)$$\sigma =E_{0}\varepsilon e^{cw}e^{\left[-{\left(\frac{F}{a}\right)}^{m}\right]}+\eta \frac{d\varepsilon }{dt}$$
where *d_σ_*/*d_ε_* = 0 at the peak stress (*σ_D_*, *ε_D_*). The derivation of ε on both sides of Formula (21) is expressed as follows:(22)$$a=\varepsilon_{D}m^{\frac{1}{m}}$$
(23)$$m=-ln\frac{\sigma_{D}-\eta \frac{d\varepsilon }{dt}}{E_{0}e^{cw}\varepsilon_{D}}$$

Wang et al. indicated that during the actual loading, the material undergoes a compaction process in the initial loading stage, resulting in permanent deformation [36]. This phenomenon leads to the failure of the damage constitutive model in the compaction stage; thus, reflecting the initial stress and strain processes in rock materials becomes difficult. Herein, the expression pertaining to the stress–strain curve that indicates the initial compaction process of the cemented backfill is established in Formula (24) through the experimental data, and the traditional constitutive is enhanced using the piecewise function as follows:(24)$$\sigma =b*(c^{\varepsilon }-1)$$

In the preceding formula, *b* and *c* denote the compaction constants related to the water content of the filling body. The statistical damage constitutive model of hydration–impact cemented backfill with a modified compaction section is obtained by sorting Formulas (21) and (24) as follows:(25)$$\{\begin{array}{l} \sigma =b*(c^{\varepsilon }-1)\begin{array}{cc} & 0<\varepsilon \leq \varepsilon_{a} \end{array}\\ \sigma =E_{0}\varepsilon_{2}e^{cw}{}^{\left[-{\left(\frac{F}{m}\right)}^{m}\right]}+\eta \frac{d\varepsilon_{2}}{dt}\begin{array}{cc} & \varepsilon_{2}\geq \varepsilon_{b} \end{array}\\ \varepsilon_{2}=\varepsilon -\varepsilon_{a}+\varepsilon_{b} \end{array}$$

Parameter $\varepsilon_{a}$ denotes the strain at endpoint A of the real stress–strain curve (Figure 6), and $\varepsilon_{b}$ denotes the strain when the ordinate of the theoretical stress–strain curve calculated from Formula (21) is equal to that of endpoint A. To unify the theoretical constitutive model with Formula (21), $\varepsilon_{2}$ can be defined as a linear function of the strain ε which connects the compaction stage and the theoretical constitutive stress–strain curve.

### 4.3. Verification of the Constitutive Model

The stress–strain theoretical curves of cemented backfill with different ω values under uniaxial dynamic compression were calculated using Formula (25) and compared with the actual stress–strain curves. The results are shown in Figure 11.

From the model verification results, it can be observed that for the ω cemented filling specimens with different water contents, the curves obtained based on the utilized model are basically consistent with the real curves of the test. The curves obtained using the proposed model can more optimally reflect the dynamic mechanical properties of the cemented filling body, such as the peak stress strength and elastic modulus, which are affected by the water content state. It exhibits an apparent rate-effect feedback and can serve the deep-well mining project more effectively. Apparently, the proposed model still exhibits some limitations in the post-peak failure stage, and fails to reflect the double peak characteristics more optimally. Therefore, subsequent research should be enhanced.

### 4.4. Damage Characteristics

Figure 12 depicts the damage evolution curve of the filling body samples under different water contents. In addition to the difference in value, the change trend is similar: the damage initially develops smoothly with strain, gradually increases, and finally tends to be gentle. The change trend exhibits a typical “S” shape. The cemented filling bodies with different water contents exhibited similar damage processes under impact loads, and the damage value of the cemented filling body increased gradually until it attained the maximum value of 1 with increasing strain. 

If the stress–strain and damage–strain curves of the cemented backfill are considered, the damage evolution mechanism of the backfill exhibited four stages. Because the stress intensity of the filling body in the initial damage stage (stage I) was low, initial defects, such as the tiny cracks of the cemented filling body, were compacted. Therefore, the damage value corresponding to the specimen in this stage was almost zero, which also reveals that the influence of water content (ω) on specimen damage can be ignored. In the stage pertaining to the stable development of damage (stage II), the damage evolution curve of the filling body specimen in Figure 12 indicates that with the continuous increase in strain, the specimen began to suffer damage, which maintained a stable upward trend. In the accelerated damage stage (stage III), the damage curve of the cemented filling body exhibits a convex trend, the damage rapidly increases, and the damage accumulation under strain increases. Compared with the change law of the energy curve pertaining to the six groups of specimens in different states, the dry specimens first entered the accelerated damage stage, and with the increase in water content, the rate of specimens entering this stage was lower. In the damage failure stage (stage IV), when the stress attained the peak value, although the strain continued to increase, the damage rate gradually decreased before stabilizing. Meanwhile, the damage value attained a value of 1, and the cemented filling body was destroyed. Generally, the damage–strain curves pertaining to the six groups of specimens with different ω values in this stage exhibited some differences, and the increase rate of the damage curve decreased with the increase in ω: the damage failure rate slowed. 

The aforementioned conclusions indicate that the damage evolution characteristics of the filling body are strongly related to its water content, which mainly affects the accelerated damage and failure stage. In the actual deep-well operation, the influence of groundwater change in underground operation should be exhaustively considered in the evaluation pertaining to the stability of the filling body.

## 5. Conclusions

The energy evolution effect and deformation damage characteristics of cemented backfill in different water-bearing states (0, 20%, 40%, 60%, 80%, and 100%) under dynamic load are deeply discussed and analyzed; thus, the actual working conditions of deep-well cemented backfill in different water-rich environments are replicated using an indoor dynamic uniaxial impact test, and the following conclusions are obtained:

(1) With the increase in water content, the σd of cemented backfill decreases from 5.03 Mpa to 1.79 Mpa. However, with the increase in water content, the micro-cracks inside the specimen will be evenly and rapidly generated, which can effectively disperse the stress; thus, the sudden explosion after energy accumulation and direct failure can be avoided, and the ductility of the backfill specimen can be increased.

(2) Under the dynamic load action, when the state of the specimen changes from dry to saturated, the fractal dimensions of the cemented filling body are 2.39, 2.45, 2.49, 2.52, and 2.61, respectively: with the gradual increase in water content, the fractal dimension growth rate tends to slow down. Meanwhile, the filling body specimens under different states mainly exhibit the tensile failure mode.

(3) According to the energy evolution law, when the strain increases, the total strain energy of the cemented backfill increases proportionally, and the elastic strain energy peaks and rapidly decreases. Different water states lead to large differences between the yield and failure stages in the stress–strain curve of the backfill specimen. When the state of the specimen changes from dry to water-saturated, the increase rate of the dissipation energy curve will be significantly reduced.

(4) The dynamic damage constitutive model of the cemented backfill is proposed based on the fitting equation of the dynamic water damage variable and compaction section. The results indicate that the proposed theoretical model can optimally reflect the characteristics of peak stress and the backfill strain under the effects of varying water contents, and the accuracy and rationality of the model are verified.

(5) The relationship between the stress and damage and strain curves is obtained. It is observed that the damage evolution of the cemented backfill under impact load exhibits a typical ‘*S*’ curve, which develops smoothly-rapidly increases-tends to be gentle. Due to the increase in moisture content, the growth rate pertaining to the damage curve of the filling body specimen gradually decreases.

## Figures and Tables

**Figure 1 materials-16-05677-f001:**
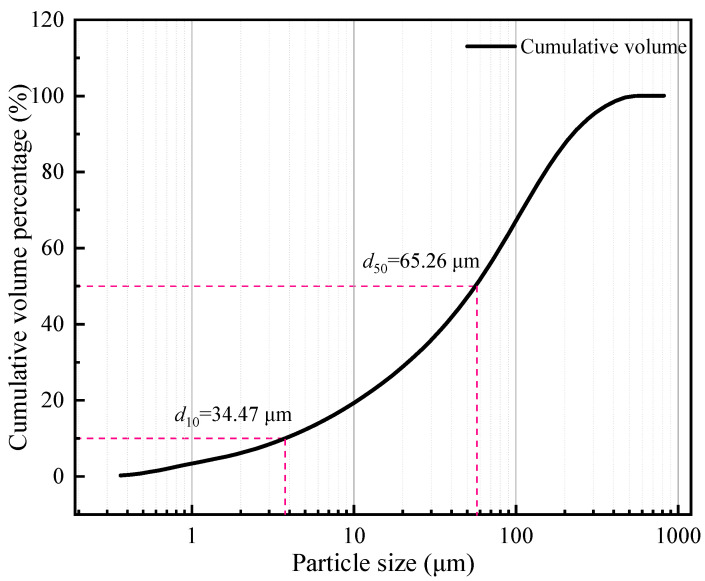
Chemical composition of graded tailings.

**Figure 2 materials-16-05677-f002:**
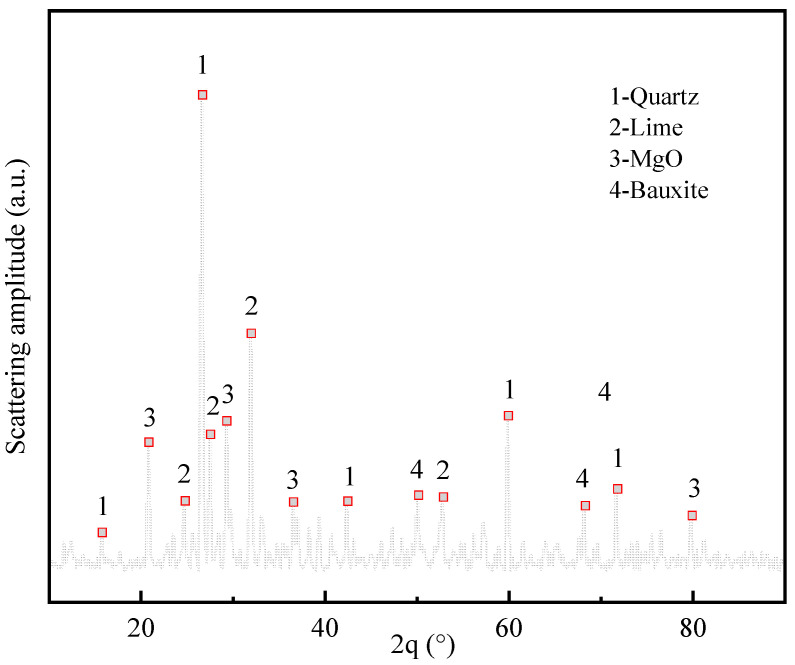
XRD patterns of tested tailings.

**Figure 3 materials-16-05677-f003:**
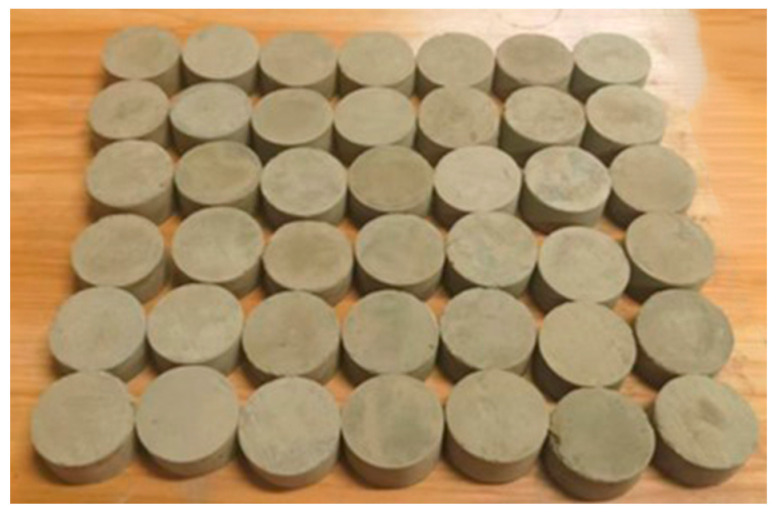
Components of cemented filling samples after processing.

**Figure 4 materials-16-05677-f004:**
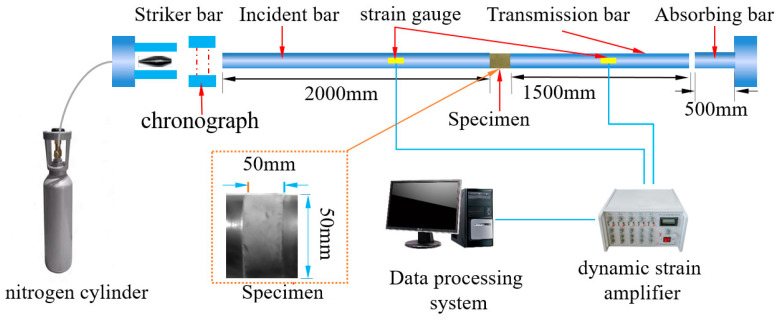
SHPB testing system.

**Figure 5 materials-16-05677-f005:**
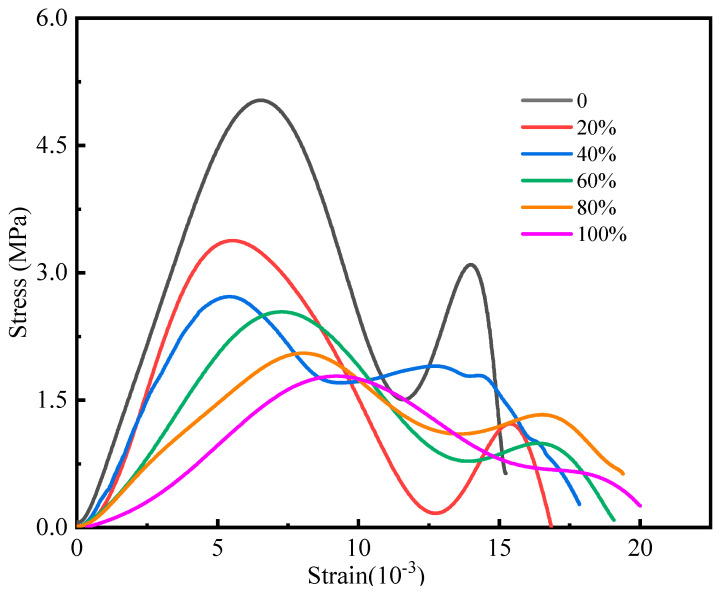
Six groups of stress–strain curves of cemented backfill in water-bearing state.

**Figure 6 materials-16-05677-f006:**
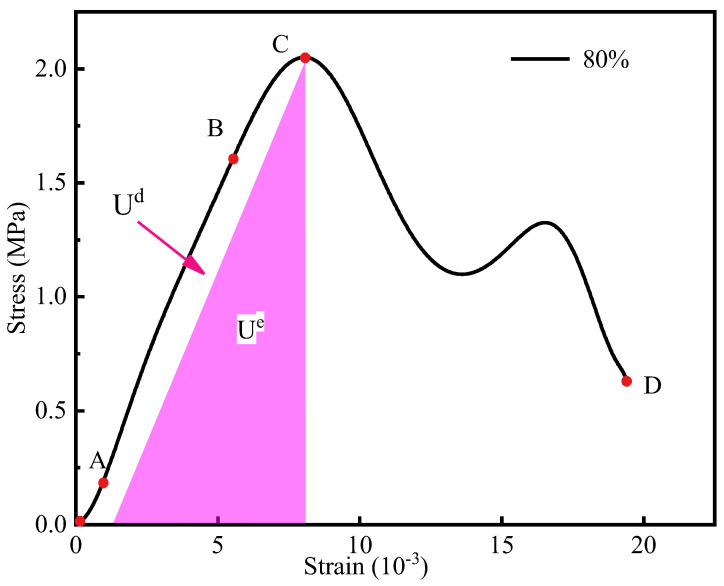
Dynamic stress–strain curve at ω = 80%.

**Figure 7 materials-16-05677-f007:**
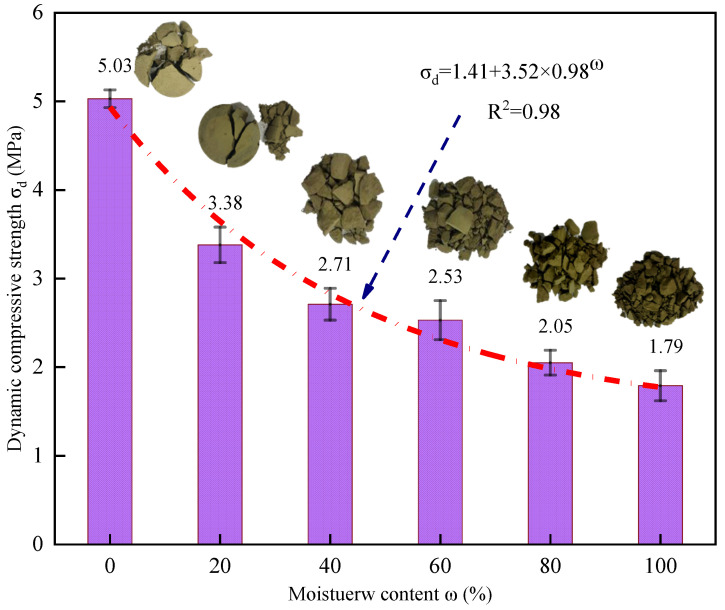
Curves of dynamic compressive strength stress with water content.

**Figure 8 materials-16-05677-f008:**
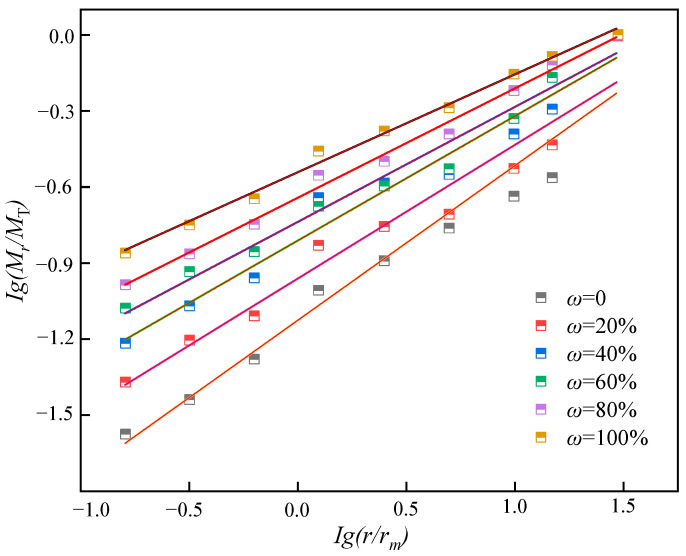
The fragmentation distribution of specimens under different water contents.

**Figure 9 materials-16-05677-f009:**
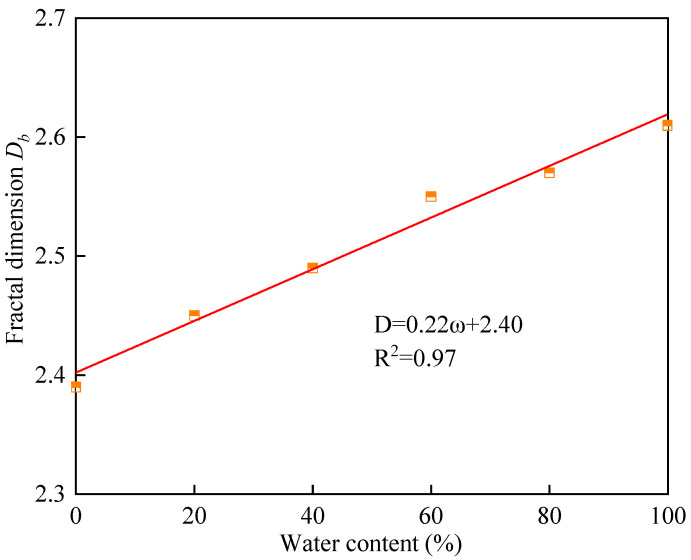
Fitting curve of water content ω and fractal dimension *D_b_*.

**Figure 10 materials-16-05677-f010:**
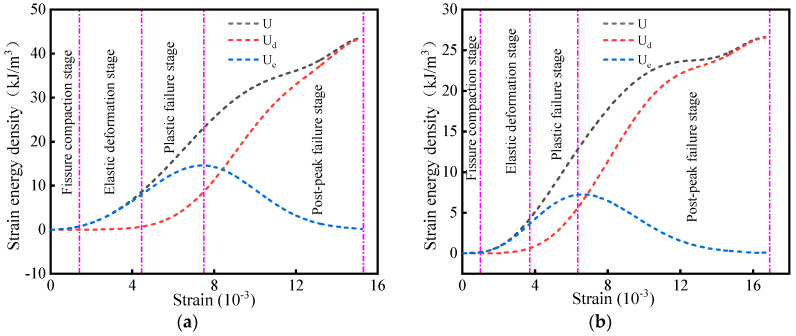
Strain–strain energy curves of specimen under different water content conditions. (**a**) ω = 0%, (**b**) ω = 20%, (**c**) ω = 40%, (**d**) ω = 60%, (**e**) ω = 80%, (**f**) ω = 100%.

**Figure 11 materials-16-05677-f011:**
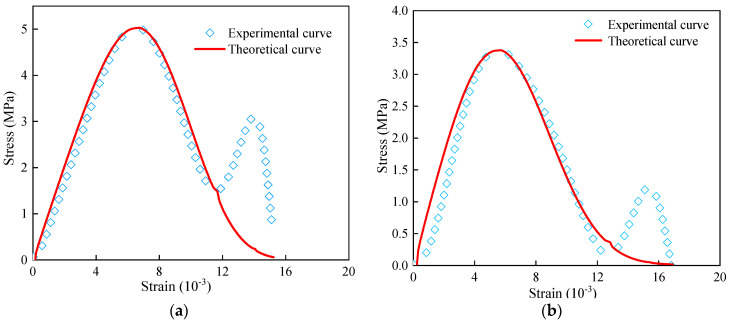
Experimental and theoretical stress–strain curves of cemented filling body. (**a**) ω = 0%, (**b**) ω = 20%, (**c**) ω = 40%, (**d**) ω = 60%, (**e**) ω = 80%, (**f**) ω = 100%.

**Figure 12 materials-16-05677-f012:**
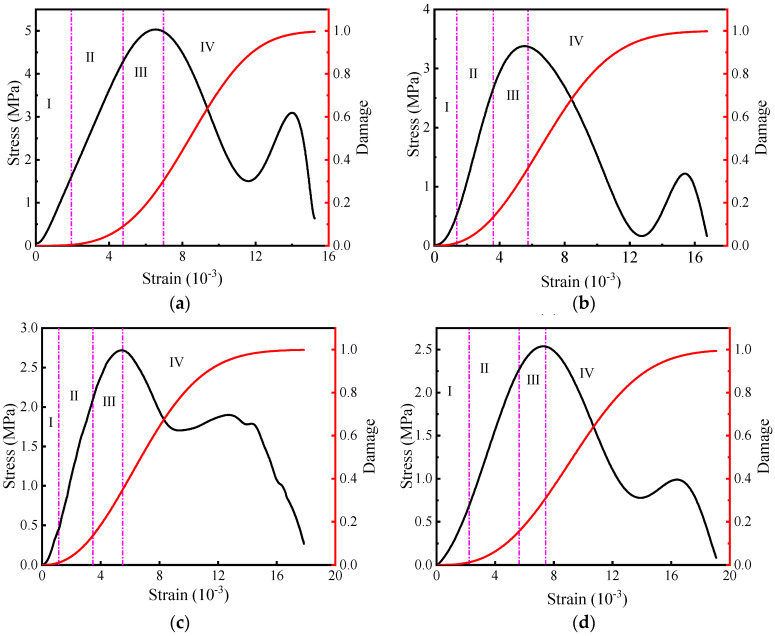
Damage evolution curves of filling sample. (**a**) ω = 0%, (**b**) ω = 20%, (**c**) ω = 40%, (**d**) ω = 60%, (**e**) ω = 80%, (**f**) ω = 100%.

**Table 1 materials-16-05677-t001:** Chemical composition of graded tailings.

Chemical Composition	CaO	SiO_2_	SO_3_	CaF_2_	Al_2_O_3_	Fe_2_O_3_	MgO
Content (wt%)	15.06	37.15	4.32	0.67	7.28	2.39	5.31

**Table 2 materials-16-05677-t002:** Curve fitting results of lg (*M_r_*/*M*_T_)-lg(*r*/*r_m_*).

Water Content ω	Fit Equation	Correlation *R*^2^	Fractal Dimension *D_b_*
0	lg(*M_r_*/*M*_T_) = 0.61lg(*r*/*r_m_*)	0.99	2.39
20%	lg(*M_r_*/*M*_T_) = 0.55lg(*r*/*r_m_*)	0.91	2.45
40%	lg(*M_r_*/*M*_T_) = 0.51lg(*r*/*r_m_*)	0.96	2.49
60%	lg(*M_r_*/*M*_T_) = 0.45lg(*r*/*r_m_*)	0.98	2.55
80%	lg(*M_r_*/*M*_T_) = 0.43lg(*r*/*r_m_*)	0.95	2.57
100%	lg(*M_r_*/*M*_T_) = 0.39lg(*r*/*r_m_*)	0.94	2.61

## Data Availability

The data used to support the findings of this study are included within the article.

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
