# Peer review of "Analyzing the Energy and Damage Constitutive of Cemented Backfill with Different Water Content under Dynamic Load"

_materials, 2023, doi:10.3390/ma16165677_

Round 1

Reviewer 1 Report

The authors have prepared six groups of cemented backfill specimens with varying water contents. The specimens are prepared to capture the actual working conditions of deep well-cemented backfill under different water-rich environments and subjected to dynamic uniaxial impact tests. Authors have investigated the effects of water content on the mechanical properties, fractal dimension, and deformation damage characteristics of cemented backfill under dynamic load. The authors demonstrate that as the state of the specimen changes from dry to water-saturated, the dynamic compressive strength and brittleness of the cemented filling body decrease significantly, and the filling-body specimens with different water contents mainly undergo tensile failure. Also, the authors have established a statistical damage constitutive model of cemented backfill.

Ultimately this work could be suitable for publication in the Materials Journal, following major revisions.

Major comments:

1.      The context of the study is not provided in the introduction. It is suggested to make a considerable improvement to the introduction. The literature review looks shallow. Add the quantitative data in the literature.

2.      Emphasise how the study filled a knowledge gap or addressed a specific issue.

3.      In the results and discussion part add more reasoning and explanations

4.      In the results, the author(s) is expected to try as much as possible to compare their results obtained with the existing or similar studies.

Minor, but still important comments that should be addressed:

1.      Abstract needs to be improved significantly by adding quantitative data.

2.      Add error bar graph for Dynamic compressive strength in Figure 7.

3.      Kindly reconcile the conclusion with the study objectives.

4.      What are the practical implications of this study and the future directions? kindly state?

The article must undergo language proofreading, as I found grammatical and typo errors. Some of the typo errors are as follows:

1.      “2.2. Specimen preparati”  Correct the spelling of preparation.

2.      Correct the spelling of “specimen” in Figure 4.

3.      Correct the spelling of “strain” in Figure 6.

4.      “Moistuerw content” Remove “w” in Figure 7 (X-axis).

5.      In figure 9, first letter of X and Y axis labels must be capital.

6.      In Figure 10, Correct the SI unit of strain energy density as kJ/m3

The article must undergo language proofreading, as I found grammatical and typo errors. 

Reviewer 2 Report

The manuscript «Study on dynamic characteristics and constitutive model of cemented backfill under different humidity conditions» by Yu Hu, Zhuo Li, Yawen Su, Yongbo Wu, Xiaoshuai Li, Wenxue Gao and Xiaojun Zhang was submitted for peer review.
I read the submitted manuscript with great interest. The authors turned to an interesting problem: study of backfill characteristics under different humidity. From my point of view, the authors did not quite correctly choose the research methodology: mining conditions are not supported, the recipe and method of preparing the material are not described. The manuscript addresses an interesting topic that has potential for application in mining.
Despite of the well-conducted study, the authors have failed to prove the relevance of the study. The manuscript has significant flaws that need to be corrected. Correction of the shortcomings listed below must be done to improve the quality of the manuscript, enhance the ease of perception of the presented material and increase the interest of a readers.
1.) From my point of view, the authors did not quite define the title correctly. The manuscript is about the study of backfill under different humidity conditions. But the introduction is about little study of the backfill characteristics under different water inflows into the mine. Since humidity and water inflow are different factors, it is necessary to bring it to a common meaning.
2.) From my point of view, this number of keywords is very few. In addition, keywords should be more direct and related to the content of the manuscript. Avoid abbreviations. Exceptions are established expressions, such as GPRS.
Keywords enable the reader to quickly search for the necessary material and enable the author to popularize their research and increase interest and citations. But if this number of keywords satisfies the requirement of the journal, this comment is advisory.
3.) The abstract is not quite formed correctly. It is very blurry and framed incorrectly. It seems that the authors have taken certain phrases from the text and thus formed the abstract. The abstract should clearly indicate the purpose of the study, its importance for society (i.e. to characterize the problem), identify the methods and materials of the study, and the conclusions should be clearly and briefly formulated. There is no "starting point" in the abstract, that is, information about previous studies (one sentence is enough). From my point of view, in the abstract, such information begins with the statement: "Previously conducted studies have established that ...".
3.1) It is desirable to avoid narrative text in the abstract.
3.2) Try to use words and phrases: an analysis has been carried out; studied; developed; proposed; established and so on. It is advisable to start sentences in the abstract with these words and phrases.
3.3) At the end of the abstract, it is necessary to indicate the final result obtained by the authors, for example: A model has been developed that allows ...; A dependence has been established which is...; A pattern has been revealed...; An efficient system (technology) has been proposed, and so on.
The abstract should be revised.
4.) The manuscript has a sufficient list of references (30 references in total). The authors mainly rely on the research of one scientific school. There is no comprehensive coverage of research in terms of geography of citations. I counted only 5 papers of international research, with all of them older than 5 years. No references to international studies in the field, especially on the work of Eastern European, Ukrainian, and Russian scientists.
The list of references is intended to demonstrate the depth of the authors’ study of the material, the relevance and interest of their research.
4.1.) Depth of study is demonstrated with the number of references - is sufficient.
4.2.) Relevance – with the availability of research in recent years – is sufficient.
4.3.) Interest – with the availability of research by scientists from different countries - is not sufficient.
I ask the authors to take this recommendation seriously. Since you are publishing your manuscript in an international publication, it is necessary to demonstrate the international relevance and interest of this issue. This can be done by analyzing the studies of scientists from different countries. It is imperative to supplement the list of references with studies of scientists from Eastern European countries over the past 3-5 years to show geographical (general/global) interest and relevance.
Major revision of References might be sufficient if these tests have been performed. Otherwise, the paper should be considered as rejected in the present form.
Below I present a few papers relevant to this study that could greatly improve the manuscript. The authors have the right to use the material proposed or offer their own versions of international studies to increase the geography of citation.
The list of references should be supplemented.
5.) In the introduction when analyzing previous studies, the authors make inaccuracies or provide information that overloads the text and often their claims are not accompanied with evidence. It is important for readers to know the essence (main idea) of the research you are referring to when analyzing previous work.
In the introduction, it is necessary to analyze the previously completed work and note what has been done, what are the shortcomings, and what has been done incorrectly. Such shortcomings are present throughout the Introduction. Authors need to revise the introduction, adjust, and supplement their statements with evidence.
5.1) I am not a native speaker, but nevertheless, in my opining, the authors form a very long sentences, which are very difficult to perceive. Such sentences greatly reduce the easy perception of the material.
5.2) In the introduction, the authors refer to several works and quite rightly state what is done in this study. However, the authors do not explain why this study is interesting: what has been done right or wrong, what can be learned from the study, what needs to be corrected or improved and why this research is important.
5.3) The authors use mining and processing waste to prepare the material. This is very correct from the point of view of preserving the environment. But the authors did not indicate this in the introduction. From my point of view, it is necessary to note the impact (accumulation) of waste on the environment and indicate the need for their disposal. One of such methods of disposal is the use of waste in backfill production.
5.4) Line 38 is inaccurate. The authors claim that «the filling mining method can reduce the ore dilution rate». However, this is not the case. The use of backfill reduces ore losses (increases recovery rates) but increases dilution.
5.5) The authors study strength of backfill but they do not analyze strength control options.
6.) The authors have arranged the references well. But from my point of view, the authors abuse the name of scientists when mentioning the study, for example Daraei et al. [14] and Yuan et al [15]. Reference [14], [15] is sufficient. If the reader is interested in the name of the researcher, then it is easy to refer to the references list. It is important for the reader to know the essence (main idea) of the disclosed issue, not the name of the researcher.
7.) At the end of the introduction, brief conclusion of the analytical study of earlier papers is absent. The authors did not summarize their analysis and did not identify unresolved issues. This conclusion should make it possible to characterize the actual question posed, the purpose of the study and the tasks to be solved to achieve this goal. For example: Analyzing the above, it can be noted that ... is a very topical issue. Therefore, the purpose of this study is ... and to achieve this, it is necessary to solve the following tasks: 1); 2); ... Such a conclusion allows the reader to understand the vector of the study, and the authors to correctly formulate the conclusions. It needs to be improved.
8.) Considering the comments (4) and (5), I would like to note that the authors have very poorly disclosed the main subject of the study.
The impact of industrial waste on the environment is quite large. Therefore, the issues of reducing this impact are relevant and scientists around the world are trying to minimize it. In recent years, many studies have been carried out on the study of a cement material based on industrial waste.
For example:
8.1) Kongar-Syuryun, C.B.; Aleksakhin, A.V.; Eliseeva, E.N.; Zhaglovskaya, A.V.; Klyuev, R.V.; Petrusevich, D.A. Modern Technologies Providing a Full Cycle of Geo-Resources Development. Resources 2023, 12, 50. https://doi.org/10.3390/resources12040050
The authors of this article claim that waste utilization without prior extraction of a valuable component, is atavistic. The paper proposes shifting geotechnology development from simple mineral extraction towards technologies that provide a full cycle of geo-resources development. A radical way of ensuring a full cycle of geo-resources development is the involvement of sub-standard ores and industrial waste in a closed processing cycle. The utilization of industrial waste without a valuable component extracting or reducing a harmful component to a background value is palliative. A comparative description of various technologies that allow extracting valuable components from sub-standard ores and industrial waste is made. The paper proposes a variant of chemical–physical technology that makes it possible to extract a valuable component from industrial waste to a minimum value.
The authors of the manuscript will be able to eliminate remark (5.3) if they analyse this article.
8.2) Kongar-Syuryun, Ch.B.; Faradzhov, V.V.; Tyulyaeva, Yu.S.; Khayrutdinov, A.M. Effect of activating treatment of halite flotation waste in backfill mixture preparation. Mining Informational and Analytical Bulletin 2021, 2021(1), 43–57. https://doi.org/10.25018/0236-1493-2021-1-0-43-57.
8.3) Kongar-Syuryun, Ch.; Tyulyaeva, Y.; Khairutdinov, A.; Kowalik, T. Industrial waste in concrete mixtures for construction of underground structures and minerals extraction. IOP Conference Series: Materials Science and Engineering 2020, 869(3), 032004. https://doi.org/10.1088/1757-899X/869/3/032004.
Papers (8.2) and (8.3) suggest activation treatment of tailings before mixing to improve the strength and rheological characteristics. Activation treatment or activating additive is one of the ways to improve the quality of the created material.
From my point of view, the studies (8.2) and (8.3) will suit the authors in the analysis of previously completed works to demonstrate various options for controlling the characteristics of the created material. In this way the authors will avoid shortcomings in recommendation (5.5).
If the authors become familiar with the works presented in (8.1), (8.2), (8.3) they will be able to properly form the introduction, enrich their manuscript with international research by scientists from Poland, Czech Republic, Slovenia, Slovakia, Russia, Germany and demonstrate the depth of their material, as well as eliminate the remarks (4) and (5).
9) Of particular interest to me, and I think readers as well, is SEM and X-RAY analysis for studying composition of tails. From my point of view, the next point should be indicated:
9.1) equipment used for studies (brand/model);
9.2) database for X-RAY analysis;
9.3) Method of sample preparation;
9.3.1) whether grinding was carried out, if so, to what fraction;
9.3.2) whether drying was carried out, if so, what the initial and final humidity was;
9.3.3) equipment for drying.
10.) Method for determining density of tails. If this information is borrowed, then indicate the source.
11.) Type of tails: current or stale tails.
12.) Description of experiment for material creating with following information:
12.1) brand of Portland cement and preferably cement manufacturer;
12.2) a standard composition (with a cement binder). If there was no such standard, the reason for this should be stated;
12.3) how was the convergence of the results achieved;
12.4) how was the homogenization (mixing) carried out; what is the mixing tool; what is the mixing velocity and time;
12.5) what is the sequence of filling of the components;
12.6) how the homogeneity of the composition (thoroughness of mixing) was achieved, provided that the amount of some components (for example, fibre) in the composite is minimal;
12.7) how were underground (mine) conditions achieved during the hardening of backfill;
12.8) composition (recipe): binder/ aggregate/activator (fibres)/ grouting fluid;
12.9) number of samples.
To eliminate remarks (9) – (10), I would recommend reading the work: Ermolovich, E.A.; Ivannikov, A.L.; Khayrutdinov, M.M.; Kongar-Syuryun, C.B.; Tyulyaeva, Y.S. Creation of a Nanomodified Backfill Based on the Waste from Enrichment of Water-Soluble Ores. Materials 2022, 15, 3689. https://doi.org/10.3390/ma15103689. The recommended paper is similar to the one submitted for peer review and describes the methodology in sufficient details.
13.) From my point of view, the study is detached from mining for the following reasons:
13.1) methodology for studying the normative characteristics of backfill was violated, namely, the period of testing the samples for compression.
13.2) mining conditions were violated at the time of curing samples
13.3) Samples are initially dried to complete curing, then irrigated with humidity, which does not happen in mining conditions. In the mine, curing occurs with the constant presence of humidity.
Summary: The manuscript is not a finished research work. The corrections are needed. The chosen research topic is relevant. From my point of view, the authors failed to present their research correctly and clearly, which reduced its value and worsened the ease of perception of the material presented. From my point of view, the manuscript cannot be published in the open press without correction in accordance with my suggestions.

Round 2

Reviewer 1 Report

The authors have addressed all the queries. The article may be accepted in its present form. 

Reviewer 2 Report

The manuscript «Analyzing the energy and damage constitutive of cemented backfill with different water content under dynamic load» by Yu Hu, Zhuo Li, Yawen Su, Yongbo Wu, Xiaoshuai Li, Wenxue Gao, Xiaojun Zhang and Genzhong Wang was submitted for second review.
As can be seen from the submitted manuscript and the explanatory note to the review, the authors did a lot of work to make changes in accordance with the comments. The revised manuscript is a completed scientific study on a highly relevant topic: study of backfill characteristics under different humidity. The revised version of the manuscript, in my opinion, fully satisfies the requirements of a scientific article and can be published in the open press.